# Deregulated Transcriptome as a Platform for Adrenal Huntington’s Disease-Related Pathology

**DOI:** 10.3390/ijms25042176

**Published:** 2024-02-11

**Authors:** Anna Olechnowicz, Małgorzata Blatkiewicz, Karol Jopek, Mark Isalan, Michal Mielcarek, Marcin Rucinski

**Affiliations:** 1Department of Histology and Embryology, Poznan University of Medical Sciences, 61-701 Poznan, Poland; anna.olechnowicz@student.ump.edu.pl (A.O.); mblatkiewicz@ump.edu.pl (M.B.); karoljopek@ump.edu.pl (K.J.); 2Doctoral School, Poznan University of Medical Sciences, 60-812 Poznan, Poland; 3Department of Life Sciences, Imperial College London, Exhibition Road, London SW7 2AZ, UK; m.isalan@imperial.ac.uk (M.I.); mielcarekml@gmail.com (M.M.); 4Imperial College Centre for Synthetic Biology, Imperial College London, London SW7 2AZ, UK

**Keywords:** Huntington’s disease, transcriptional deregulation, adrenal glands, mouse model

## Abstract

Huntington’s disease (HD) is a neurodegenerative disorder that affects mainly the central nervous system (CNS) by inducing progressive deterioration in both its structure and function. In recent years, there has been growing interest in the impact of HD on peripheral tissue function. Herein, we used the R6/2 mouse model of HD to investigate the influence of the disease on adrenal gland functioning. A transcriptomic analysis conducted using a well-established quantitative method, an Affymetrix array, revealed changes in gene expression in the R6/2 model compared to genetic background controls. For the first time, we identified disruptions in cholesterol and sterol metabolism, blood coagulation, and xenobiotic metabolism in HD adrenal glands. This study showed that the disrupted expression of these genes may contribute to the underlying mechanisms of Huntington’s disease. Our findings may contribute to developing a better understanding of Huntington’s disease progression and aid in the development of novel diagnostic or therapeutic approaches.

## 1. Introduction

Huntington’s disease (HD) was first described over 150 years ago and has been characterized by impaired mobility and behavioral changes [1,2]. Currently, HD prevalence ranges from 0.4 in the Asian population to 5.7 per 100,000 in the populations of Europe, North America, and Oceania [3].

One of the characteristic features of HD is a loss of control over movements, which worsens over time and with the progression of the disease [2]. Another symptom includes depression [4]. HD is caused by CAG expansion within the huntingtin gene (*HTT*) located on chromosome 4 [5,6]. Individuals with HD are characterized by 40 or more CAG repeats, where having 36–39 CAG repeats may lead to incomplete penetration of the disease [6,7,8,9,10].

Considering the widespread expression of *HTT* in virtually all tissues [11,12], it is reasonable to infer that symptoms or functional changes related to HD can also manifest in multiple organs [13,14]. Research conducted by Bruyn et al. revealed reduced plasma levels of dehydroepiandrosterone sulfate (DHEA-S) in HD patients, suggesting potential alterations in adrenal functions associated with HD [15]. Similar findings were obtained by Leblhuber et al., who observed lower DHEA-S serum levels in HD patients compared to healthy controls [16]. In the context of cortisol levels, Bruyn et al. demonstrated low cortisol serum levels in HD patients; however, this could be explained by diurnal variations in cortisol levels [15]. In contrast, research by Heuser et al. [17] and Leblhuber et al. [16] indicated elevated cortisol plasma levels in HD patients. Additionally, a study by Björkqvist et al. [18] showed elevated cortisol levels in urine samples from HD patients compared to healthy individuals. Heuser et al. [17] reported elevated plasma levels of adrenocorticotropic hormone (ACTH) in HD patients compared to healthy controls, demonstrating dysregulation of the hypothalamic–pituitary–adrenal (HPA) axis.

Animal studies have contributed valuable insights into HPA axis malfunction in Huntington’s disease. Transgenic R6/2 mice expressing human exon 1 of the huntingtin gene with approximately 150 CAG repeats [19] exhibited notable adrenal gland enlargement at 7 weeks of age compared to wild-type controls [18]. Furthermore, a detailed morphometrical analysis revealed that these observed alterations occurred in the adrenal cortex. In addition to structural changes, the R6/2 mouse model exhibited distinct hormonal variation. Compared to the wild type, these mice showed elevated serum levels of corticosterone starting from 5.5 weeks of age and ACTH at 12 weeks of age, accompanied by lower corticotrophin-releasing hormone (CRH) levels at 12 weeks of age [18].

Similar studies using the transgenic R6/1 mouse model, expressing exon 1 of the human huntingtin gene with approximately 115 CAG repeats [19], focused on catecholamine production by the adrenal glands [20]. In this model, the levels of noradrenaline, adrenaline, and dopamine were found to be lower compared to those in wild-type controls. These observations provide further evidence of adrenal dysfunction in HD mouse models. Cumulative data from both human and animal studies underscore the intricate impact of the aberrant form of huntingtin on the HPA axis and adrenal glands in the context of Huntington’s disease pathology.

Our research sought to extend the understanding of Huntington’s disease (HD) beyond its impact on the nervous system by examining transcriptomic alterations in adrenal glands of the R6/2 mouse model. Hence, our study focused on elucidating the transcriptomic profile of adrenal glands in the symptomatic R6/2 mice and comparing it to genetic background controls.

A microarray analysis, along with application of various bioinformatic tools, revealed predominant down-regulation of gene expression in the adrenal glands of R6/2 mice. Notably, a substantial portion of those down-regulated genes were associated with metabolic processes in the adrenal glands. 

Our research thus contributes to a growing body of evidence indicating that HD affects crucial metabolic processes in peripheral organs such as the adrenal glands. These insights deepen our understanding of the systemic nature of HD pathology, providing a foundation for further investigations into the molecular mechanisms linking mutant huntingtin to metabolic dysregulation.

## 2. Results

The microarray analyses were performed on adrenal glands from fully symptomatic 9-week-old R6/2 mice in comparison to genetic background controls. In our analysis, the adrenals from the R6/2 mice exhibit increased expression of 19 genes and decreased expression of 105 genes compared to the genetic background controls (Figure 1A). Differentially expressed genes were determined using the following cut-off criteria: abs (fold change) > 2 and *p* < 0.05 (with 10% FDR correction). Furthermore, we conducted principal component analysis (PCA) to visualize the first and second components of the filtered microarray dataset. The first principal component accounts for 45.2% of the total variance, while the second principal component accounts for 21.9% of the variance (Figure 1B).

Our initial analysis concentrated on a subset of the most dysregulated genes in the adrenal glands of the R6/2 mice as compared to the adrenal glands from the genetic background control. The tabulated data in Figure 1C highlight the ten most up-regulated and down-regulated genes, presenting their symbols, gene names, fold changes, and corresponding *p*-values. Notably, the gene with the highest fold change by 7.57-fold is fatty acid 2-hydroxylase (Fa2h). Conversely, the gene with the most significant down-regulation is serine (or cysteine) peptidase inhibitor, clade A, member 1D (Serpina1d), by −17.41-fold. 

In order to demonstrate which processes can be regulated by differentially expressed genes, we conducted an enrichment analysis of the ontology groups using the DAVID GO BP DIRECT database. Our analysis shows 49 biological processes related to down-regulated genes in the R6/2 group compared to the genetic background controls (Figure 2A). The processes that involved most of the altered genes are: the exogenous drug catabolic process, the xenobiotic metabolic process, and the lipid metabolic process. Additionally, by using the DAVID KEGG PATHWAY database (Figure 2B), we identified fourteen down-regulated pathways in HD mice compared to the genetic background control. The most altered number of genes was assigned to “Metabolic pathways” (33 genes), while the lowest number was categorized into “Phenylalanine metabolism” (4 genes). Similarly to the DAVID GO BP DIRECT ontology terms analysis, no KEGG pathway in this database was found to be significantly enriched by up-regulated genes.

To further explore the transcriptomic differences observed in the adrenals from the R6/2 mouse model, we conducted analyses using the GeneTonic library [21,22]. This allowed us to perform analogous enrichment analysis of the ontological groups, as presented above, but with reference to different reference databases. The enriched biological terms were clustered using Ward’s method, with color-coded branches representing different ontological clusters (Figure 2C). We found new down-regulated processes, such as a response to alcohol, a defense response to bacterium, and a response to metal ions. Fibrinolysis, the negative regulation of endopeptidase activity, the response to metal ions, and the negative regulation of peptidase activity are characterized by very high z-scores (Appendix A). To determine changes in the expression of genes belonging to the analyzed sets, we further studied 15 gene sets characterized by the lowest *p*-value. The heatmap shows the set of genes with altered expression involved in each process (Figure 2D). The majority of these genes are also present in heatmaps of four clusters related to the different processes (Appendix A). Fibrinolysis and the negative regulation of endopeptidase activity (both containing nine genes each) are the processes with the highest number of down-regulated genes.

To further analyze the processes altered in the R6/2 adrenals, significantly enriched ontological terms from DAVID GO TERM DIRECT were grouped into four clusters based on distinct characteristics (referring to their function). These clusters included processes related to the following biological classes: (i) cholesterol metabolism, (ii) blood coagulation, (iii) steroids and catecholamines, and (iv) xenobiotic metabolism (Appendix A). Genes influencing the enrichment of processes within each biological class were subjected to hierarchical clustering and are presented as heatmaps along with their corresponding fold change values. The cluster of processes involved in cholesterol metabolism includes a total of 11 processes (Appendix A). The lipid metabolic process has the highest number of genes (13), while the processes with the fewest genes (3 each) are the lipoprotein biosynthetic process and the positive regulation of cholesterol metabolism. *Apoa1* is implicated in all listed events except for lipid hydroxylation. *Ces1g* and *Lcat* are the second most altered genes, participating in seven processes each. The genes with the most reduced adrenal expression in the R6/2 group compared to the genetic background control group are *Cyp3a41b* (fold −16.2), *Cyp3a41a* (fold −16), and *Apob* (fold −8.9). Conversely, the genes with the least reduced expression are *Ehhadh* (fold −2.1), *Lcat* (fold −2.2), and *Cyp2e1* (fold −2.4). The cluster related to steroids and catecholamines encompasses a total of three ontological terms (Appendix A). The GO term with the most up-regulated genes (12) is the steroid metabolic process. Among these three enlisted GO terms, none of them are assigned to all terms. The genes with the most reduced expression in the R6/2 group compared to the genetic background control group are *Cyp3a41b* (fold −16.2) and *Cyp3a41a* (fold −16), which are the same genes involved in processes related to cholesterol metabolism. *Slc18a1* is the gene with the least reduced expression (fold −2) in the R6/2 group. The ontological terms associated with blood coagulation form a cluster including eight processes (Appendix A). Those events that contain the highest number of deregulated genes are as follows: blood coagulation (nine genes), fibrinolysis (seven genes), and the down-regulation of fibrinolysis (six genes). *Fga*, *Fgb*, and *Fgg* are genes involved in six ontological terms each. Those genes with the most reduced expression in the R6/2 group compared to the genetic background control group are *Fgb* (fold −8.5), *Fgg* (fold −8), and *Plg* (fold −6.4). *Dbh* is the gene with the least reduced expression (fold −2.3). The fourth analyzed cluster is related to xenobiotic metabolism and contains four processes (Appendix A). The GO terms with the highest number of genes involved (13) are the xenobiotic metabolic process. Similarly to the cluster associated with catecholamines, we observed no gene participating in all four GO terms. Notably, *Cyp2b9* is the only gene involved in three of these terms. The genes with the most reduced expression in the R6/2 group compared to the genetic background control group are *Cyp3a41b* (fold −16.2), *Cyp3a41a* (fold −16), and *Cyp2b9* (fold −9.1). The genes with the least reduced expression are *Chrna3* (fold −2.1), *Cyp2e1* (fold −2.4), and *Slc6a4* (fold −2.5).

To analyze gene sets that were either enriched or depleted in the adrenals of the R6/2 mouse model compared to the genetic background control, we performed gene set enrichment analysis (GSEA). For all genes, we calculated an enrichment score (ES) for each gene set based on Gene Ontology (GO). Then, we calculated the normalized enrichment score (NES) based on each gene set’s size. Subsequently, we grouped gene sets from the highest to lowest NES and identified the 20 most enriched and depleted gene sets (Figure 3A). The majority of these sets are responsible for processes involved in the immune system. Additionally, we performed running enrichment score analyses for the five most enriched and five most depleted gene sets (Figure 3B,C).

Finally, we checked the expression of endogenous huntingtin and its related genes in the R6/2 model compared to the genetic background control. For this purpose, we extracted the expression of *Htt* and *Htt*-related genes from the microarray data, and compared these results obtained from the R6/2 group to the genetic background control group (Figure 4). We observed no significant changes in the expression levels of the studied genes in the adrenals from the R6/2 group compared to the genetic background control. This result indicates a lack of impact of endogenous *HTT* transcripts in the R6/2 model on the expression of these genes. At the same time, it proves that these changes in gene expression were not caused by potential changes in endogenous *Htt* or *Htt*-related gene expression but by exogenous mutant *HTT* fragments.

In order to ensure the tissue specificity of the observed gene expression changes in the adrenal glands, we conducted an analysis comparing the expression changes in adrenal differentially expressed genes (DEGs) with those in other tissues (Figure 5). Transcriptomic data derived from various tissues of the R6/2 mice and corresponding controls were retrieved from the GEO database. These findings revealed that the expression changes in the analyzed DEGs are tissue-specific and, with the exception of single genes, do not overlap with the expression profile observed in the adrenal glands of the R6/2 mice.

## 3. Discussion

As a neurodegenerative disorder, Huntington’s disease is known for the gradual deterioration of the CNS’s structure and function. The adrenal glands play a crucial role in hormone regulation and the stress response. For a broader understanding of how Huntington’s disease impacts the body’s overall function, we comprehensively analyzed the transcriptomic profile of an HD mouse model (R6/2) to indicate impairment in the adrenal-derived steroid hormone production processes. Our analysis may contribute to developing a better understanding of Huntington’s disease progression and aid in the development of novel diagnostic or therapeutic approaches. 

We revealed significant changes in gene expression in the adrenal glands of the R6/2 mouse model compared to the genetic background control. The DAVID GO BP DIRECT and KEGG PATHWAY enabled us to identify specific pathways and gene sets altered by the expression of expanded CAG repeats in exon 1 of human *HTT* related to the adrenal glands’ metabolic role. We showed an inhibitory effect of Huntington’s disease on adrenal gland biological pathways by identifying the genes and mechanisms impaired in HD based on GO BP DIRECT analysis. We found that cholesterol and sterol metabolism, blood coagulation, and xenobiotic metabolism were mostly inhibited in the R6/2 model compared to the genetic background control. GSEA analysis allowed us to investigate gene sets that are both enriched and depleted in the HD model. Additionally, the analysis of the expression of endogenous *Htt* and Htt-interacting genes showed no significant changes between the R6/2 mouse model and the genetic background control group. It confirmed that those changes in gene expression observed in different processes were not caused by the expression of endogenous huntingtin but by the presence of an exogenous human fragment of mutant *HTT*. The R6/2 mouse model expressing a fragment of human huntingtin with about 150 CAG repeats becomes symptomatic at about 6–9 weeks of age [19]. 

Herein, we confirmed that cholesterol biosynthesis and metabolism were altered in the R6/2 model of Huntington’s disease, which may disturb the production of steroid hormones. The most significant gene responsible for lipid and sterol metabolism in the adrenal glands is *Cyp17a1* [23], whose expression was diminished in the adrenal glands of the R6/2 model. Interestingly, the study performed on *Cyp17a1^−/−^* knockout mice indicated an increase in corticosterone production in a group of males but not females [24]. Moreover, studies including the R6/2 mouse model showed an enhanced serum corticosterone level and ACTH, which stimulates adrenal glands to synthesize hormones [18]. Our results are consistent with previous findings and allow us to conclude that the decrease in *Cyp17a1* expression in HD mice may be related to the increase in corticosterone levels. Moreover, the expression of the *Cyp3a41* family is highly reduced in the adrenals of the R6/2 mice (fold change: −16), which may have a crucial impact on steroid metabolism processes. Furthermore, the CYP46A1 protein has been identified as an enzyme that restricts the rate of cholesterol degradation in the brain [25] to prevent neuronal dysfunction. An additional study focusing on a link between the *Cyp17a1* and Cytochrome P450 family of adrenal hormones in the R6/2 model might be a promising path for the future research. 

The disrupted expression of *ApoB* and *Apoa1* is currently the focus of extensive analysis efforts in HD patients [26]. Several studies suggest that changes in lipid composition and lipoprotein metabolism disruptions play a role in the underlying HD pathomechanisms [26]. Our study indicates that the expression level of both apolipoproteins was suppressed (fold change: −8.9, and −5.2, respectively) in the R6/2 mouse model. Chang et al. showed that plasma levels of ApoB and ApoA1 were significantly reduced in symptomatic and presymptomatic HD patients [26]. 

Furthermore, our data indicate that *Th* (tyrosine hydroxylase) was suppressed in the R6/2 model, which is responsible for the conversion of tyrosine to dopa during dopamine synthesis (fold change: −2.5) [27]. An analysis by Dickson et al. [28] determined the transcriptome profile using microarray analysis in the hypothalamus of transgenic mice expressing the mutant *HTT* fragment. The *Th* expression level in the HD mouse model was lower compared to the control group. Consequently, it has been showed that Th protein levels decreased between 8 and 15 weeks of age in some parts of the brain of R6/2 mice [29]. Similarly to *Th*, the *Ddc* (dopa decarboxylase) expression level decreased in the adrenal glands −2.3-fold. Down-regulation of this gene was also found in the hypothalamus of an HD mouse model [28]. Meanwhile, the human ortholog of this gene was characterized by altered methylation at the presymptomatic stage in HD patients’ [30]. Dopa decarboxylase activity was elevated in the erythrocytes of HD patients compared to the control group [31]. However, Butterfield et al. suggested that changes in dopa decarboxylase activity in HD patients could be connected to phenothiazines used often in therapy [31,32]. 

*Prl* (prolactin) showed decreased expression (fold change: −6.8). Serum prolactin levels in HD exhibit gender-related variation, with women tending to display higher expression, albeit without statistical significance [33]. Moreover, we confirmed that the expression of *Otc* (ornithine transcarbamylase) was also reduced (fold change: −3.7), which is consistent with a lower expression of OTC protein in the blood in pre-symptomatic HD patients compared to controls [34].

Furthermore, we observed alterations in the expression of crucial genes for the central nervous system, *Emb* and *Fa2h*, which play essential roles in maintaining nervous system functionality. The Emb protein contributes positively to nerve terminal sprouting [35], while the Fa2h protein is implicated in central nervous system myelination [36]. The increased expression of these genes within the adrenal gland may be attributed to associated nerve fibers, where these changes potentially take place. 

We also found that *Dbh* (dopamine beta-hydroxylase), responsible for the positive regulation of vasoconstriction, was reduced in the R6/2 model (fold change: −2.3). The DBH protein converts the catecholamine dopamine to norepinephrine [37]. A growing body of evidence indicates that the expression of human DBH is enhanced in the plasma of patients with HD compared to healthy controls [38,39,40]. A study by Garrett and Soares-da-Silva [41] analyzing cerebrospinal fluid showed higher dopamine concentrations in HD patients than in healthy subjects. A study by Kaplan et al. [42] showed lower dopamine secretion in various striatal regions of R6/2 mice compared to controls. Additionally, a study by Martínez-Ramírez et al. [20] using R6/1 mice showed lower adrenal medullary dopamine levels in the symptomatic HD mice, and as well lower Dbh levels. To summarize, most studies involving HD patients reveal that their DBH levels are higher than in healthy subjects, which also correlates with higher dopamine levels. This is in contrast to a number of studies performed in HD mouse models, where the expression levels of both Dbh and dopamine were reduced. It should be noted that such a discrepancy could be caused by different Dbh levels in various tissues. 

The *F2* (coagulation factor II) expression level was reduced in the analyzed R6/2 model (fold change: −3.8). It participates in fibrinolysis, blood coagulation and platelet activation. It has been described that F2 is responsible for producing prothrombin, which is elevated in the cerebrospinal fluid in HD patients [43]. 

The analysis of *Htt* and Htt-interacting gene expression showed no significant changes between the R6/2 model and the genetic background control group. This confirms that the impaired expression of genes presented in this study was caused by the mutant *HTT*, and not by the expression of the endogenous *Htt* gene. Furthermore, an examination of alterations in tissue-specific expression profiles validated that the identified gene expression is specific to the adrenal glands. 

Despite the valuable insights gained from this study, several limitations must be acknowledged. The primary limitation of this study arises from the constrained sample size, which regrettably hindered the feasibility of performing additional molecular analysis analyses like a Western blot. These limitations impact the scope of our findings but offer opportunities for further investigation and the refinement of future research endeavors.

## 4. Materials and Methods

### 4.1. Animals

The R6/2 mouse model (B6CBA-Tg(HDexon1)62Gpb/3J) and C57BL/6J mice were obtained from Jackson Laboratories (US). Breeding of R6/2 mice was conducted as previously described [44]. Briefly, R6/2 males were bred with (CBA×C57BL/6) F1 females (B6CBAF1/OlaHsd, Harlan Olac, Bicester, UK) through backcrossing to obtain hemizygous R6/2 female mice. Genotyping to determine the number of CAG repeats in the human HTT fragment was performed as previously described [44]. The number of CAG repeats was equal to (162 ± 2.8 SD). Both R6/2 mice (n > 6) and control mice used in this study were sacrificed at 9 weeks of age. Rodents’ housing conditions were the same as previously described [45]. Briefly, animals had unlimited access to water and breeding chow (Special Diet Services, Witham, UK) and were provided with environmental enrichment. Also, mice were kept in a 12 h light-and-12 h dark cycle. All conducted experiments involving animals were carried out in accordance with the approval received from the Home Office, UK, and with approval from the Animal Welfare and Ethical Review Body of Imperial College London, project license number: PAB101FA8.

### 4.2. RNA Isolation

All animals were sacrificed by cervical dislocation, and adrenal glands were harvested immediately. Subsequently, each tissue underwent blood clearance through perfusion using cold PBS and snap-frozen. To prepare samples for microarray analysis, total RNA was extracted with the TRI Reagent (Sigma-Aldrich, St. Louis, MO, USA). We then proceeded to the purification of samples using columns (NucleoSpin Total RNA Isolation, Qiagen GmbH, Hilden, Germany). The quantity and quality of total RNA was defined using a NanoDrop spectrophotometer (Thermo Scientific, Waltham, MA, USA). In addition, the quality of the total RNA was verified by a Bioanalyzer 2100 (Agilent Technologies, Inc., Santa Clara, CA, USA) (an average RNA integrity number (RIN) of 9.2 (obtaining RINs from 8.5 to 10)). Next, samples were diluted to acquire a concentration of 100 ng/µL.

### 4.3. Microarray Study

The microarray assay was performed according to the manufacturer’s protocol, as described earlier [44,46,47,48]. The total RNA was pooled into seven samples, which comprised R6/2 mice (n = 4) and a genetic background control (n = 2). A total of 100 ng of each sample was subjected to two rounds of amplification of sense cDNA, biotin labeling, and fragmentation using the GeneChip^®^ WT Plus Reagent Kit, Affymetrix, Santa Clara, CA, USA. Hybridization of biotin-labeled cDNA fragments (5.5 µg) to the Affymetrix^®^ Mouse Gene 2.1 ST Array Strip (45 C/20 h) was then conducted. Subsequently, array strips were washed and stained using the Fluidics Station of a Gene Atlas System (Affymetrix), followed by scanning of array strips using the Imaging Station from a Gene Atlas System. Preliminary analysis was performed via Affymetrix Gene Atlas TM Operating Software (version 2.0; Affymetrix, Santa Clara, CA, USA) on the scanned arrays. The quality of the data of gene expression was examined using software quality control criteria, and CEL files were obtained for further analysis.

### 4.4. Data Analysis of Microarray Study

The analysis incorporated tabular data containing information on fold changes, adjusted *p*-values, and normalized counts for each comparison. This dataset underwent analysis via a BioConductor repository in tandem with the statistical programming language R (version 4.1.2; R Core Team 2021). The identification of differentially expressed genes (DEGs) involved specific criteria: an absolute fold change exceeding 2 and *p*-value < 0.05 corrected with a 10% false discovery rate (FDR). The normalization and computation of expression values for the analyzed genes utilized the Robust Multiarray Average (RMA) algorithm from the “Affy” library [49]. To visualize the number of up- and down-regulated genes, a volcano plot generated by the ggplot2 package was used. A principal component analysis (PCA) was performed on the on the set of genes with the highest variance and graphically represented using the “factoextra” library [50]. To explore and observe the transcriptomic differences, we used the “GeneTonic” library [21,22]. We ranked the altered processes in terms of their −log10 *p*-value and determined their z-score, which represents changes in the gene expression of the analyzed sets. For functional annotation and clustering of differentially expressed genes (DEGs), the DAVID (Database for Annotation, Visualization, and Integrated Discovery) bioinformatics tool was employed [51]. Subsequently, expressed genes were linked to pertinent Gene Ontology (GO) terms, and significantly enriched GO terms were identified via the GO BP DIRECT and KEGG pathway databases. *p*-values of the selected GO terms were corrected using the Benjamini–Hochberg method [52]. Hierarchical clustering of DEGs and their visualization as a heatmap for each comparison were performed using the “ComplexHeatmap” library [53]. 

Gene set enrichment analysis (GSEA) was performed using the cluster Profiler Bioconductor library [54]. The aim of the analysis was to identify sets enriched or depleted in Gene Ontology (GO) terms by calculating the normalized enrichment score (NES) along with the *p*-value. The normalized fold change (FC) values for all genes underwent logarithmic transformation and sorting, and were subsequently utilized as input for the gseGO function. The analysis focused on the “biological process” GO category, with the predefined minimum gene set size for analysis set at 100 and a Fisher exact *p*-value threshold of ≤0.05. A bar chart was created to illustrate the ten ontology groups with the highest enrichment scores (highest NES values) and the ten groups with the most depleted enrichment scores (lowest NES values). Additionally, enrichment plots were provided for the top five most enriched and the top five most depleted GO terms.

### 4.5. Assessment of Tissue-Specific Expression Profile Changes in Analyzed Genes

To investigate whether changes in expression profiles exhibit tissue specificity, we retrieved transcriptomic data from R6/2 mice and their corresponding controls for various organs. The data were obtained from the Gene Expression Omnibus (GEO) repository, utilizing selection conditions for expression profiling by array specific to R6/2 mice (Table 1). The tissue-specific data presented in the table below were employed for subsequent analysis. For each comparison, samples from the control and test groups were identified using GEO2R software [55]. Subsequently, the generated R code was utilized to select genes corresponding to those differentially expressed in R6/2 adrenal glands. This procedure involved merging the resulting tables based on gene symbols. Genes whose symbols were absent in at least three datasets were excluded from the final table. The outcome of this operation was a table containing fold change values. The resulting data were visualized as a heatmap, accompanied by hierarchical clustering applied to both columns and rows.

## 5. Conclusions

Our studies using the R6/2 mouse model and microarray analysis allowed us to determine the transcriptomic profile and compare the obtained findings to previous literature reports on processes taking place in HD adrenal glands. We demonstrated a number of down-regulated gene sets related to cholesterol metabolism and catecholamine biosynthesis, of which some genes and their products had already been studied in HD. The obtained results may form a basis for further studies focused on specific processes occurring in adrenal glands.

## Figures and Tables

**Figure 1 ijms-25-02176-f001:**
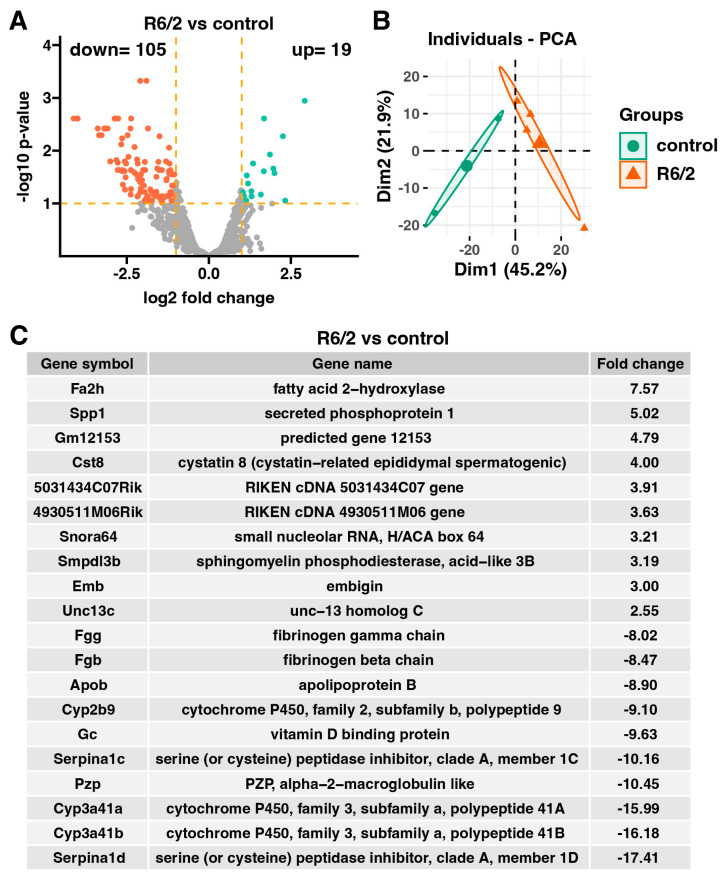
The overall gene expression profile derived from microarray analysis of the R6/2 mouse adrenal glands in relation to the genetic background control. (**A**) Volcano plot of differently expressed genes. Differentially expressed genes are separated by yellow dashed lines located at cut-off values: *p*-value < 0.05 with 10% FDR correction and |fold change| = 2. Down-regulated genes are denoted by orange dots and up-regulated genes are indicated by green dots. The counts of down- and up-regulated genes are positioned in the upper part of the plot. (**B**) Principal component analysis (PCA) of adrenal gland samples in R6/2 and genetic background control groups. The diagram presents first two components of the filtered microarray dataset. Green dots represent the genetic background control group, and orange triangles denote the R6/2 group. (**C**) List of ten most up-regulated and ten most down-regulated genes in the R6/2 group in comparison to the genetic background control group.

**Figure 2 ijms-25-02176-f002:**
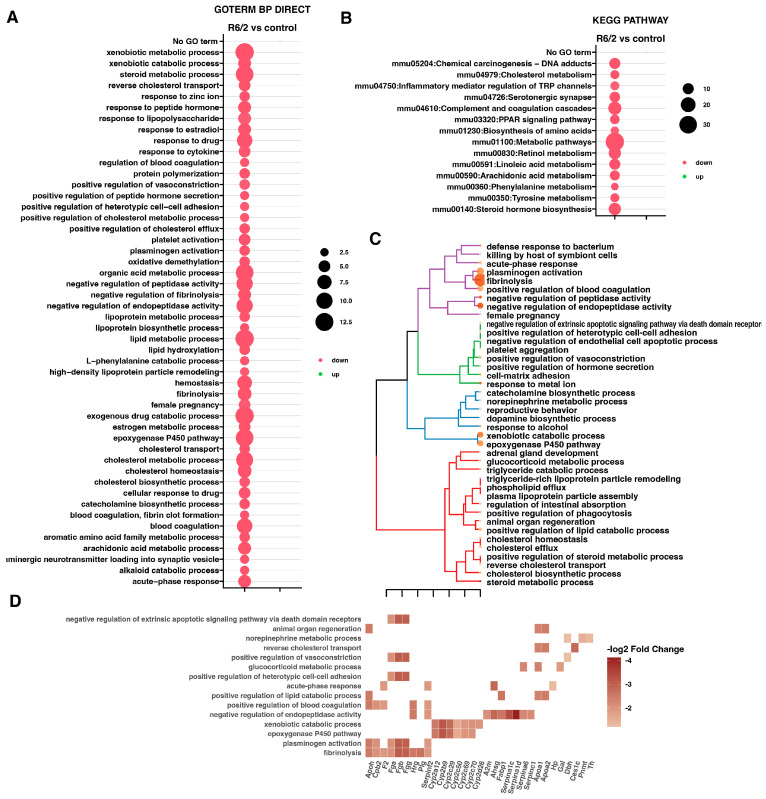
(**A**,**B**) Bubble plots representing differentially regulated gene ontology group from the DAVID GO BP DIRECT database. Columns present down-regulated gene sets represented by red bubbles. No statistically significant up-regulated ontological terms were found (green bubbles). The bubbles’ size is correlated with the number of genes in the particular gene set. The criterion used for qualification of GO terms is an adj *p*-value < 0.05. (**C**,**D**) Enrichment analysis of ontological terms using the GeneTonic library. (**C**) Dendrogram illustrating four clusters of differentially regulated gene sets. The size of dots placed by the GO term negatively correlates with the *p*-value, while the color positively correlates with the z-score. (**D**) Heat map representing differentially regulated genes from the 15 processes with the lowest *p*-values from plot C. Various shades of brown indicate a fold change.

**Figure 3 ijms-25-02176-f003:**
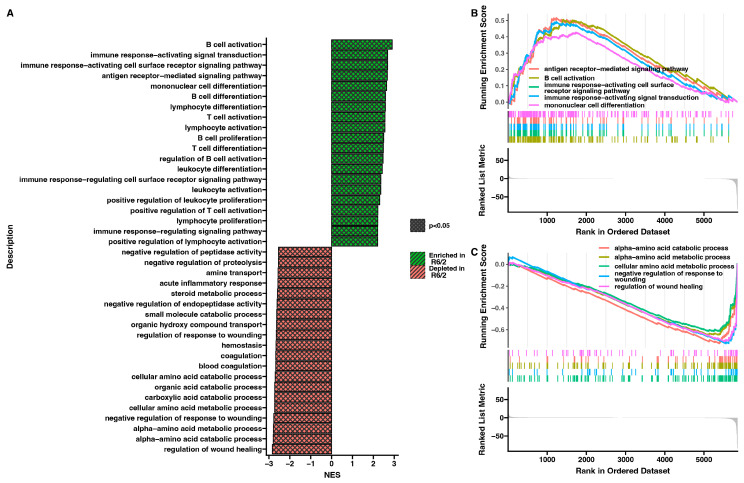
Gene set enrichment analysis (GSEA) represents gene sets differently regulated in the R6/2 group. (**A**) Bar graph representing the twenty most enriched (green) and twenty most depleted (red) gene sets in the R6/2 group based on their normalized enrichment score (NES). Black checking in the bars represents *p*-values < 0.05. (**B**) Enrichment plot of the five most enriched gene sets from graph A with their running enrichment scores. (**C**) Enrichment plot of the five most depleted gene sets from graph A with their running enrichment scores.

**Figure 4 ijms-25-02176-f004:**
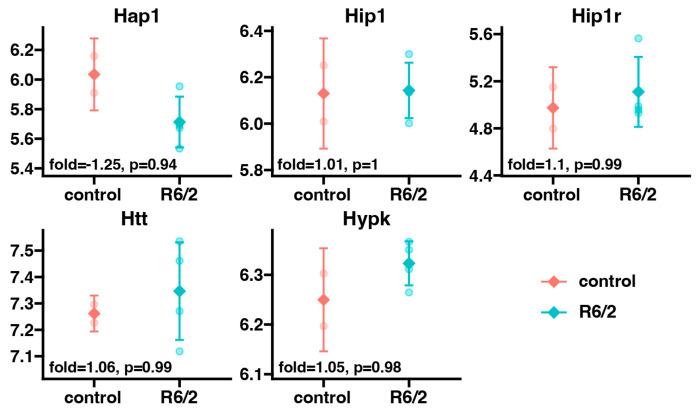
Expression of *Htt* and Htt-related genes in adrenal glands of the R6/2 group compared to the genetic background control group. The turquoise color represents the R6/2 group, and the orange color represents the genetic background control. Fold change values and *p*-values are positioned in the bottom left corner of each graph.

**Figure 5 ijms-25-02176-f005:**
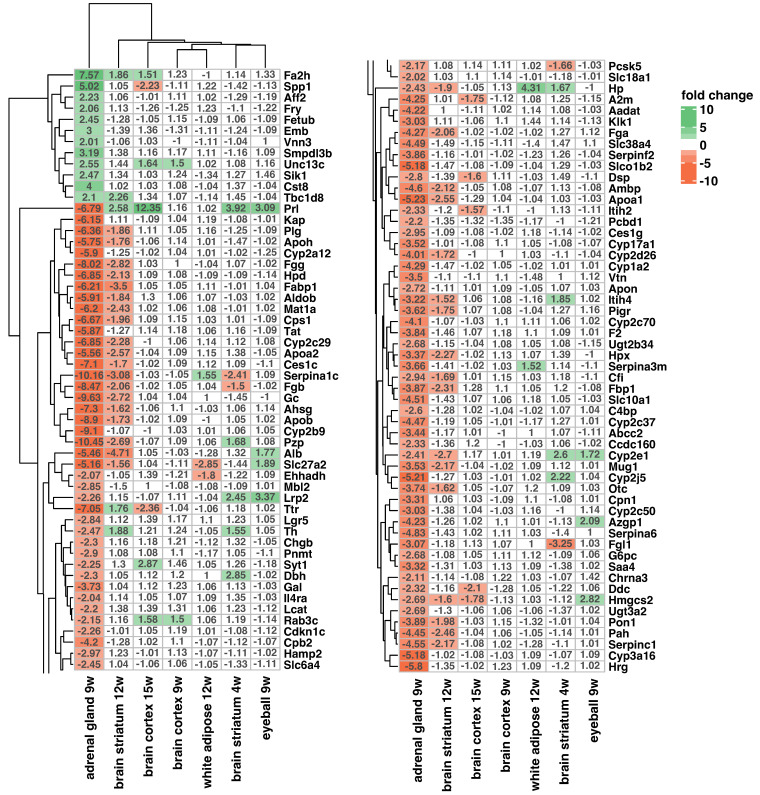
Tissue-specific expression profile changes in analyzed genes. The heatmap illustrates fold change values representing the expression changes in regulated genes across different tissues in R6/2 mice. Lifespan is indicated in weeks (w) on labels. Fold changes above and below |1.5| are color-coded. The heatmap is divided into two parts for enhanced presentation.

**Table 1 ijms-25-02176-t001:** The data used for the comparative analysis of gene expression profiles (links accessed on 22 January 2024).

Authors	GEO Accession Number	Analyzed Tissue	Compared Groups	Lifespan
Mazur-Michalek I. et al. [44]	GSE199335	eyeball	R6/2 vs. WT	9 weeks old
McCourt AC. et.al. [56]	GSE79711	subcutaneous white adipose	R6/2 vs. WT	12 weeks old
Mielcarek M. et al. [57]	GSE38219	brain cortex	R6/2 vs. WT	15 weeks old
Mielcarek M. et al. [57]	GSE38218	brain cortex	R6/2 vs. WT	9 weeks old
Labbadia J. et al. [58]	GSE29681	brain striatum	R6/2 vs. WT	12 weeks old
Miyazaki H. et al. [59]	GSE113929	brain striatum	R6/2 vs. WT	4 weeks old

## Data Availability

All the data discussed in this work, if not already included in the manuscript, are available from the corresponding author on reasonable request.

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
