# Peer review of "Deregulated Transcriptome as a Platform for Adrenal Huntington’s Disease-Related Pathology"

_ijms, 2024, doi:10.3390/ijms25042176_

Round 1
Reviewer 1 Report
Comments and Suggestions for Authors
I thank the authors for their valuable contribution to the science.
There are some grammatical errors that need to be addressed. Please take a look at the attached PDF file. Also, the authors used HTT, HD, and R6/2 throughout the manuscript. Please be consistent. To be precise, R6/2 should be used. Also, for control, use either the term genetic background control or B6. There is no wild type.
Figure 9A needs more line spacing, and letters are too small on 9B, and C. Figures 9A and 9B are duplicates. I suggest using Figure 9B and moving 9A to supplemental.
Some crucial studies have been left out and not cited. Please search gene names followed by Huntington's disease to add those references. For example, https://pubmed.ncbi.nlm.nih.gov/31286142/
Some of the studies referred to in the discussion are brain studies. The authors did mention that in the body but failed to state that the discrepancy could be due to different tissue types.
Figures 4-7 are an expansion of Figure 2. Figures 4-7 should be supplemental. The authors only used a sentence to summarize each figure, so it is unnecessary to include it in the body.

There are many grammatical errors. Need a professional help.
Author Response
We thank this reviewer for their positive comments and time devoted to improving our manuscript. We appreciate your valuable feedback and careful observation. We appreciate your diligence in reviewing our work. We have carefully considered the comments in preparing our revision. Please find the attached revised version (All changes are marked yellow). Detailed Responses to reviewers’ comments are listed below. Moreover, the article was revised by a language expert.
“There are some grammatical errors that need to be addressed. Please take a look at the attached PDF file. Also, the authors used HTT, HD, and R6/2 throughout the manuscript. Please be consistent. To be precise, R6/2 should be used. Also, for control, use either the term genetic background control or B6. There is no wild type.”
Thank you for raising this important point. We have carefully checked the entire article and made the necessary corrections according to the reviewer's suggestions.
“Figure 9A needs more line spacing, and letters are too small on 9B, and C. Figures 9A and 9B are duplicates. I suggest using Figure 9B and moving 9A to supplemental.”
Thank you for this suggestion. For a better presentation of the main findings of this study, we have reorganized the figures. Please see the revised version of the manuscript. Moreover, according to the reviewer’s suggestion, we removed some figures from the supplementary materials.
“Some crucial studies have been left out and not cited. Please search gene names followed by Huntington's disease to add those references. For example, https://pubmed.ncbi.nlm.nih.gov/31286142/”
Thank you for this comment. The reference [25] relevant to the topic, was added. Please find the revised version of the References section highlighted in yellow.
“Some of the studies referred to in the discussion are brain studies. The authors did mention that in the body but failed to state that the discrepancy could be due to different tissue types.”
Thank you for this suggestion. We improved the discussion section, please find the attached version of the manuscript.
“Figures 4-7 are an expansion of Figure 2. Figures 4-7 should be supplemental. The authors only used a sentence to summarize each figure, so it is unnecessary to include it in the body.”
Thank you for raising those issues. We removed Figures 4-7 from the supplementary material and rewrote the results section.
Reviewer 2 Report
Comments and Suggestions for Authors
The manuscript from Olechnowicz et al. evaluated the transcriptomic landscape of the R6/2 the adrenal glands. The authors used Affimetrix’ microarray assay and evaluated gene expression differences between 3 WT and 4 R6/2 mice.
Authors used classical GO term and pathway enrichment analysis besides GSEA to identify mostly downregulated pathways in the adrenal of this HD mouse line.
The manuscript is well written and properly structured, but can be shortened and represented in a more concise manner.
I suggest that the manuscript is shortened in terms of figures, for example merge PCA, differential expression analysis and pathway enrichment analysis in one single figure.
Similar for figures 4 to 7.
The study lacks biological relevance because there is no validation of any proposed target. Authors make claims such as “we confirmed that cholesterol biosynthesis and metabolism are altered in R6/2 model of HD” lines 316-317. I don’t think this is a correct statement, because the authors only detected decreased expression of genes related to these processes, which doesn’t directly imply that these pathways are altered.
There are infinite possibilities to generate figures from molecular profiling experiments, such as the presented microarray. I think the authors should adequate the number of figures and length of the results/discussion to the reality of the manuscript, that is just one single experiment.
If the authors do not validate any of the proposed targets, they at least should try to integrate the current transcriptomic data to some of studies previously published.
Author Response
We thank this reviewer for their positive comments and time devoted to improving our manuscript. We appreciate your valuable feedback and careful observation. We appreciate your diligence in reviewing our work. We have carefully considered the comments in preparing our revision. Please find the attached revised version (All changes are marked yellow). Detailed Responses to reviewers’ comments are listed below. Moreover, the article was revised by a language expert.
“The manuscript is well written and properly structured, but can be shortened and represented in a more concise manner.
I suggest that the manuscript is shortened in terms of figures, for example merge PCA, differential expression analysis and pathway enrichment analysis in one single figure.
Similar for figures 4 to 7. “
Thank you for raising this point. We improved the manuscript to make it more consistent.
“The study lacks biological relevance because there is no validation of any proposed target. Authors make claims such as “we confirmed that cholesterol biosynthesis and metabolism are altered in R6/2 model of HD” lines 316-317. I don’t think this is a correct statement, because the authors only detected decreased expression of genes related to these processes, which doesn’t directly imply that these pathways are altered.
There are infinite possibilities to generate figures from molecular profiling experiments, such as the presented microarray. I think the authors should adequate the number of figures and length of the results/discussion to the reality of the manuscript, that is just one single experiment.
If the authors do not validate any of the proposed targets, they at least should try to integrate the current transcriptomic data to some of studies previously published.”
Thank you for this suggestion. The article has been carefully revised to make it more consistent. For a better presentation of the main findings of this study, we have reorganized figures. Please see the revised version of the manuscript. Moreover, according to the reviewer’s suggestion, we removed some figures from the supplementary materials. To confirm the specific of the revealed findings, we performed a comprehensive analysis based on publicly available data from the GEO database. We indicated that the examination of alterations in tissue-specific expression profiles validated that the identified gene expression is specific to the adrenal glands. We also discussed limitation of our study by pointing out that further proteomic analysis would be needed to validate transcriptomic alterations in HD adrenal glands.
Round 2
Reviewer 2 Report
Comments and Suggestions for Authors
I accept the paper in the present revised form